# Toward secure mobile applications through proper authentication mechanisms

**Abdulmohsen Saud Albesher** [1]*, **Amal Alkhaldi**[1], **Ahmed Aljughaiman**[2]

**1** Department of Information Systems, College of Computer Sciences and Information Technology, King Faisal University, Al-Ahsa, Saudi Arabia, **2** Department of Computer Networks and Communications, College of Computer Sciences and Information Technology, King Faisal University, Al-Ahsa, Saudi Arabia

\* aalbesher@kfu.edu.sa

## Abstract

With the increased number of mobile apps, authentication processes play a key role in verifying users' identities and protecting data from security threats. Utilizing proper authentication techniques is key to protecting computer apps from being hacked. In this paper, we aimed to compare the authentication methods of the sign-up, sign-in, and password recovery processes of 50 e-commerce apps. To ensure accurate data analysis, we checked every app in a separate session and used the "think-aloud" technique while recording the screen. The researchers prepared a list of items that were checked during each session to identify the similarities and differences between tested apps regarding the authentication process. The results of this security analysis unequivocally demonstrated how different apps' designs for authentication processes are. Users' memory and comprehension are burdened by these variances, and no app can ensure that they adhere to recommended standards. The results of this study confirmed the necessity for unified and user-friendly authentication processes. This can be possible by following a usable security framework for the authentication process.

**Data Availability Statement:** https://doi.org/10.6084/m9.figshare.27759330.v1.

## Section 1: Introduction

Nowadays, protecting mobile apps such as government, banking, healthcare, and social media apps is necessary because they contain users' personal and sensitive information. Authentication in mobile apps plays a significant role in protecting users' personal data on their smartphones and preventing unauthorized access to the phone. User authentication has effectively become one of the biggest challenges facing mobile apps today [1].

The variety of app purposes and their ability to accomplish most of people's tasks has led to them accessing sensitive information such as financial and health records [2]. To protect users' personal information from being compromised, the authentication processes on smartphones should be elevated to high levels of security. Thus, it is important to study the current techniques used in these processes. App authentication needs to be highly usable without compromising security. Recently, techniques of fingerprint, voice, and face recognition have helped to improve usability because they help fast access and provide greater convenience during the login process.

**Funding:** The authors extend their appreciation to the Deanship of Scientific Research, Vice Presidency for Graduate Studies and Scientific Research, King Faisal University, Saudi Arabia, under Grant KFU242550. The funders had no role in study design, data collection and analysis, decision to publish, or preparation of the manuscript.

**Competing interests:** The authors have declared that no competing interests exist.

**Abbreviations:** Abbreviation, Definition; CMAF, Chaotic Map-based Authentication Framework; NIST, National Institute of Standards and Technology; DSR, Design Science Research; OTPs, One-Time Passwords; DT, Decision Tree; PCs, Personal Computers; FHE, Fully Homomorphic Encryption; PGaaS, Password Guessability as a Service; GAN, Generative Adversarial Network; PMs, Password Managers; GPS, Global Positioning System; SaaS, Security as a Service; IBM, International Business Machines; SMDs, Smart Miniature Devices; IIoT, Industrial Internet of Things; SSL, Secure Sockets Layer; ML, Machine Learning; TLS, Transport Layer Security.

Mobile apps can interact with about 80% of the world's population, with 6.4 billion smartphones in use globally [3]. Due to the increasing number of smartphone users, developers must enhance the usability and security of their apps. Many of the security problems people face today in mobile apps, such as security breaches and data theft, are caused by security vulnerabilities and weak user authentication processes [4]. Security is an important attribute that should be taken into consideration while developing apps.

One of the key security mechanisms is the authentication process. Most smartphone and mobile apps apply user authentication techniques to verify the user's identity before allowing them to perform any additional operations. Furthermore, using these devices has become a necessity not a choice for individuals; apps have become an essential part of an individual's life. Smartphone apps have entered many fields such as online shopping, internet banking, navigation, and social media [5]. This has encouraged developers to increase the usability and security of their apps by using the best authentication practices to protect personal information. As a result, users should feel more comfortable dealing with the authentication process.

Smartphones use different authentication techniques to verify users' identities and protect data from security threats. This paper focuses on evaluating the authentication practices in 50 apps. We selected 50 apps from the App Store. The apps exist in one domain (i.e., e-commerce domain). The apps were chosen based on their "Similarweb" ranking. Every app was tested individually by observing several items during three processes (signup, sign-in, and password recovery). Examples of these items include the types of password policies and password meters, the necessity of retyping passwords, the ability to hide or show passwords, and using biometrics and password managers.

Albesher [6] compared the authentication procedures for websites and indicated that there is a need to study the procedures for mobile apps. The results of this study should help developers to choose the most appropriate authentication practices for e-commerce apps to ensure high usability and security. The main contributions of this paper are as follows:

1. Compare the authentication mechanisms for mobile apps.

2. Identify the similarities and differences between these authentication mechanisms.

3. Explore the most common authentication mechanisms for mobile apps.

The security of user authentication mechanisms is paramount in mobile e-commerce apps, as these apps frequently handle sensitive information such as credit card numbers, personal data, and credentials for financial accounts. Despite the high stakes, there is a lack of comprehensive studies examining the real-world authentication practices employed by mobile e-commerce apps. This research provides a large-scale comparative analysis of the authentication mechanisms implemented by 50 popular mobile eCommerce apps. Our findings offer insights into the prevalence of various authentication strategies, the adoption of recommended security practices, and the detection of vulnerabilities in the authentication logic of these apps. The significance of this research is threefold. First, it provides system designers and developers with a benchmark understanding of the current state of authentication mechanisms in mobile e-commerce apps, highlighting effective practices and potentially risky design choices. Second, the research identifies prevalent vulnerabilities that could be exploited by attackers, informing the prioritization of security patches and updates. Finally, our comparative approach allows for the analysis of how different e-commerce apps influence the chosen authentication mechanisms, providing a nuanced understanding of the authentication landscape in mobile e-commerce. By shedding light on the strengths and weaknesses of authentication in mobile e-commerce, this research aims to guide improvements in the security and privacy protections

afforded to users when interacting with these increasingly popular platforms for online transactions.

The structure of this paper is as follows. Section 2 provides a brief background of authentication and mobile phones. Section 3 shows the related work associated with authentication mechanisms, password policies, password meters, and password management. Section 4 explains the methodology used to conduct this research. Section 5 provides the results, while Section 6 presents the discussions based on the given results. Finally, Section 7 highlights some future directions while Section 8 provides a conclusion.

## Section 2: Background

Authentication is considered one of the key fundamental aspects of protecting mobile apps from cyber threats. Mobile apps authenticate users through either single factor or multifactor authentication. Authentication can be broadly grouped into three categories: something the user knows, something the user has, and something user is, as illustrated in Fig 1 [2, 5, 7, 8]. Authentication began with the use of passwords in the 1960s when the first computers were made available to the general public. Computers were massive, extremely expensive, and slow compared to today's standards. The fact that only a few universities and large-scale businesses owned a computer shows how inaccessible computers were in the past. Computer users were often researchers and students who used them to calculate and store their research and new findings. Although all computer users had access to everything within the computer, users were not concerned about losing their data.

Fernando Corbató provided the password mechanism solution to access a resource in 1961. Corbató developed a simple password program where computer users could save their passwords in a plain text file on the file system. Storing passwords in plain text files is risky because they can be easily leaked [9]. Mistakes in the past pushed developers to come up with another

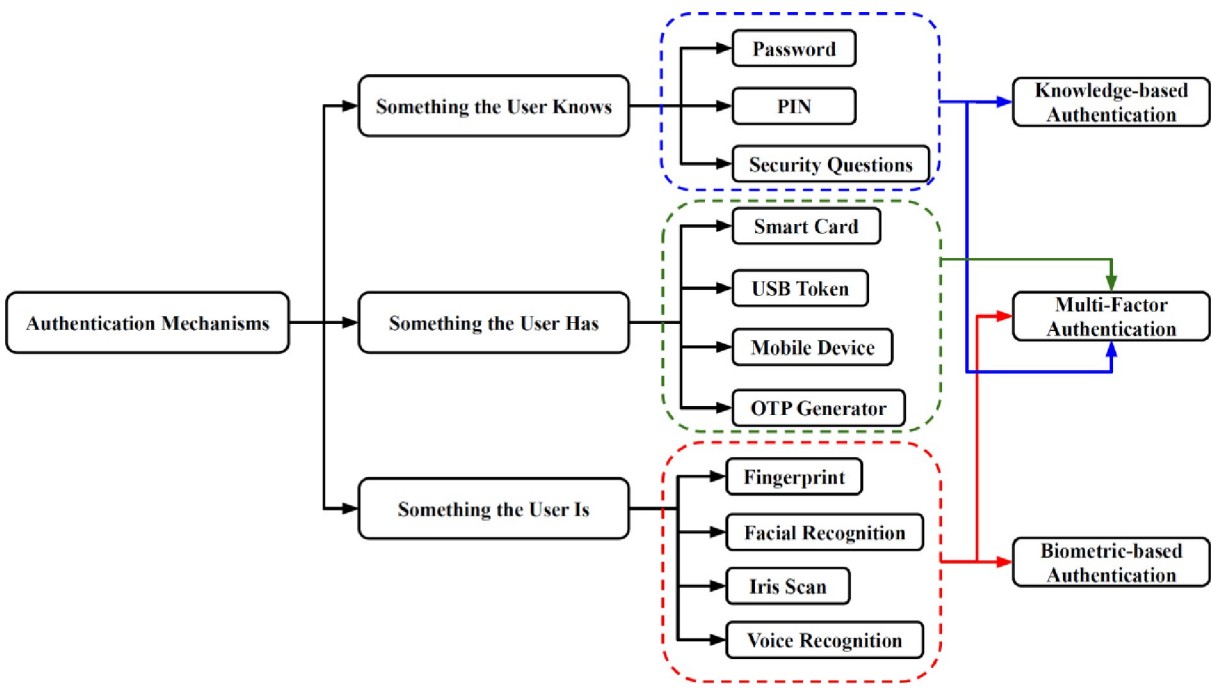

**Fig 1. Authentication mechanisms.**

authentication mechanism to protect computer systems from being compromised. Robert Morrison borrowed a concept from cryptography known as "the hash function." At the time, hash-based password storage was truly revolutionary; the hash function provided some security because it protects a user's sensitive information and ensures data integrity [10]. However, hackers discovered a way around hashing when it became a standard. Hashed passwords are also vulnerable to parallel cracking [10]. Aside from hashing, additional cryptography techniques proved useful for the advancement of authentication.

Asymmetric cryptography is a method of encrypting and decrypting data using a public and a private key. The public key may be shared among several users, but the private key is kept secret to a specific user.

As security measures have become more robust, so have the hackers. This requires the security industry to constantly step up its game and look for more secure authentication solutions, such as the idea of one-time passwords (OTPs) in the 1980s. In the 1990s, public key infrastructure went public. This resulted in the creation of the SP4 protocol, which was later renamed to the more familiar Transport Layer Security (TLS) protocol.

A few years later, the Secure Sockets Layer (SSL) protocol was created, which included server authentication and keys. The rise of multi-factor authentication and single sign-on began in the 2000s, and around this time the need for a specific hardware device to generate dynamic passwords was eliminated. Mobile apps are specialized for multi-factor authentication, where users receive OTPs that have emerged as a result of emerging technologies and digitalization. With the rise of biometrics in the early 2010s, fingerprint scanning, facial recognition, voice patterning, iris scanning, and other biometric authentication methods are now used [11, 12].

## A. Mobile app authentication

Several scientific and national reports, such as those produced by the US Department of Homeland Security, the National Institute of Standards and Technology (NIST), and the International Business Machines (IBM) global analysis report, state that many security breaches occur due to human factors [13, 14]. Further studies show that millions of users around the globe select easy to guess passwords such as "123456," the same as their username, slang, or a common dictionary word [1, 2, 8, 15]. This increases the risk of compromising users' accounts and, hence, risks mobile apps being compromised by malicious users [1, 13, 16, 17]. Fundamentally, to address these risks, password policies must be defined properly to secure mobile apps during the sign up, sign in, and recovery processes while ensuring usability.

Cell phones and tablets have been expanded widely. Nowadays, a single user may have more than one cell phone and a tablet at home. These devices allow users to download mobile apps. The number of mobile apps available in the major app stores is over two million [18]. Mobile devices and tablets often contain many apps, which increases the risk of compromising cell phones or tablets due to improper authentication mechanisms. Mobile app authentication enforces one of the common authentication mechanisms in three different areas: signup, sign in, and password recovery. In the signup phase, users must register to create an account. To set the users' credentials, there are two common approaches: either the mobile app allows users to create their passwords while signing up, based on given criteria, or the app generates passwords for new users and shares them through the user's email. In the sign in phase, users are requested to provide their credentials to access their accounts. In the recovery phase, users request to reset their passwords so that they can access their accounts. In all three phases, the program developer needs to ensure that the password policies, password meter, and password manager, among other parameters, are enforced properly.

## B. Mobile phones

Cell phones have evolved since their inception when they were initially only used to send and receive calls and messages. However, due to rapid changes in technology, today's cell phones allow users to take photos, check emails, search the Internet, and use built-in global positioning system (GPS) services, among other functions. Since some features on cell phones are not available on personal computers (PCs), these features often mean that cell phones have higher capabilities than PCs. For example, most cell phones allow users to share their device's location, which can be used to provide accurate directions and traffic updates, personal safety, service delivery, and emergency services, among other benefits [5]. In the event of a natural disaster, mobile government (m-government) can take advantage of the locations of residents so that it can send early notifications to them. As a result, information delivery in m-government is more efficient as citizens can be easily reached.

Cell phone technologies have been developed in three areas: hardware, software, and networks. In terms of hardware, modern phones have significantly improved in terms of storage and battery performance in comparison to legacy cell phones. Regarding data storage, modern cell phones have increased storage capacity, support advanced storage technology, integrate with cloud services, and feature expandable storage. In terms of battery performance, modern cell phones have increased battery capacity, utilize advanced battery technologies, enable fast charging, support wireless charging, and utilize energy-efficient components. Similarly, modern cell phone software has also been improved, including operating systems and mobile apps. Correspondingly, cell phone networks have been improved significantly in the past few years [16].

## Section 3: Related work

### A. Authentication mechanisms

Authentication can be used to protect and secure apps in smartphones and tablets from being compromised. Many researchers have investigated authentication issues to ensure mobile apps are secure from common cyber-attacks.

Kružíková [19] investigated user authentication in online banking, open-source platforms, IT security policy awareness and compliance, and user perspectives on smartphone security to identify proper authentication mechanisms. The study aimed to ensure usable security authentication in multiple forms by studying modern authentication methods and assisting system developers and security policy writers to enhance usable security. The author relied on quantitative (e.g., questionnaires, logs) and qualitative (e.g., interviews) approaches to collect data. The author collected 89 responses from the questionnaire and met 502 smartphone users to evaluate four authentication methods on smartphones. The findings of this research helped to improve users' perceptions and evaluations and facilitate work with proper authentication schemes in the online banking environment.

Kunda and Chishimba [20] studied mobile security threats and authentication protocols on Android devices. The authors analyzed several authentication schemes used on smartphones to identify trade-offs. In particular, the authors studied the following authentication schemes: password, pattern, fingerprint, facial, vocal, and iris. The results of this study indicated that biometric schemes are generally more secure than traditional authentication methods mostly due to the fact that the metrics used in biometric authentication cannot easily be replicated. Although traditional schemes offer lower security in comparison to biometric schemes, they cost less and are easy to implement and maintain. This means that biometric schemes can offer higher security but require higher costs and computational complexities.

Brown et al. [1] studied a more secure, transparent, and convenient authentication mechanism. This research introduced a novel multimodal authentication system that relies on machine learning (ML) and blockchain. The proposed system combined multiple authentication schemes, including fingerprint, face, age, and gender, to be authenticated. The proposed system has been developed and uses the decision tree (DT) algorithm for the verification process to ensure a high confidence level. The experiment was carried out by six users, and each user was required to check more than 100 biometric samples. The samples were graded from poor to excellent. The results of this study show that the system behaves as expected and produces a high confidence level when it relies on excellent samples. In contrast, the system behaved unexpectedly and produced a low confidence level when it relied on poor samples.

Ashibani et al. [18] proposed a user authentication model based on mobile application access interactions and identification methods by extracting multiple features to identify users more accurately. The authors deployed the proposed model in a real system to evaluate user authentication methods and minimize error rates when identifying users. The results of this study show that several user interactions can be used to differentiate between users and, hence, can be used to identify users more accurately. The authors used two datasets to evaluate the proposed model in terms of false positive, false negative, and equal error rate. Based on the statistical analysis of the extracted features and the results, it proved that utilizing mobile app interactions can lead to high accuracy with minimal error rates. Moreover, the proposed model promises great results in smart home scenarios. This indicates that the proposed model can assist administrators in limiting unauthorized access to systems, which increases security and privacy while ensuring usability.

Horcher [14] designed a virtual keyboard based on common usable security principles to ensure that small devices are authenticated with strong passwords. To achieve this requirement, security input efficiency and effectiveness must be considered. To evaluate and design the virtual keyboard optimally, this research used the design science research (DSR) methodology. The design of the virtual keyboard focuses on layout, size, and common user behaviors. The virtual keyboard has been evaluated by collecting each dimension of usability according to the International Organization for Standardization (ISO) standard. To ensure the effectiveness of the designed virtual keyboard, efficiency, effectiveness, and user satisfaction have been measured. Current mobile devices fail to meet the required standards of these ISO criteria; however, the results of Horcher's paper show that the designed virtual keyboard can meet all ISO criteria and, hence, improve usable security. Users can benefit from improved usable security by minimizing cognitive effort and potential security threats while using the virtual keyboard.

Tanveer et al. [21] introduced the Chaotic Map-based Authentication Framework (CMAF) which is an authentication framework designed for the Industrial Internet of Things (IIoT). It combines chaotic maps with the resource-efficient authenticated encryption scheme to address the unique security challenges of the IIoT environment. CMAF-IIoT ensures reliable communication between smart devices (SMDs) and users by establishing a secure session key after mutual authentication. With this key, users can access real-time SMD information securely. The authors validated CMAF-IIoT's security through rigorous formal and informal analyses. Their evaluation showed that compared to other frameworks, CMAF-IIoT offers low storage, computational, and communication costs, making it a viable and efficient solution for the IIoT.

## B. Password policies

A password policy is a set of rules designed to enhance app security by requiring users to employ strong passwords and use them properly. Password policies are considered the first

defense method that assists administrators in preventing unauthorized access to user accounts. To maintain security and privacy, administrators must develop strong password policies so that users' passwords are difficult to guess or crack [22]. There are several standards and best practices that can be utilized to develop password policies according to business needs; however, many top websites fail to meet common standards and best practices [22]. Effective rules can force users to change their passwords regularly and require that their passwords meet the defined level of password complexity. Strong password policies can minimize security risks related to password breaches. To have effective password policies, administrators must design password policies in a way that does not accept common, easily guessed passwords, allows the selection of complex passwords, and utilizes strong password meters to assist users in selecting strong passwords. When developing password policies, administrators must define length requirements, specify allowed characters, and define prohibited characters. Without meeting password policy requirements, users are unable to complete their registration or reset their passwords.

Password policy development must pass through a lifecycle to ensure they are comprehensive, effective, and up to date [22–30]. First, administrators must assess and plan for password policy development based on risk assessment, collecting requirements (regulatory, industry standards, and organization), and involving critical stakeholders to meet their requirements and address concerns. Second, the policy is designed by setting the objectives, writing the draft policy, and reviewing the draft with stakeholders to ensure that the draft meets the requirements and is sufficient. Third, the password policy must obtain formal approval, and there should be a communication plan to discuss the new policy with all users and training and awareness sessions to ensure users understand the new policy. Fourth, the new policy is implemented by configuring the system, deploying the policy across the organization based on the communication plan, and providing help and proper mechanisms to assist users in deploying the new policy. Fifth, continuous monitoring of password usage and compliance is conducted, and enforcement mechanisms are employed to ensure users comply with given policies. Sixth, administrators need to periodically review policies and evaluate their effectiveness and relevance, collect feedback from users and stakeholders to address any concerns, and analyze security incidents to identify areas that require further improvement. Seventh, policies should be revised and updated as necessary. Eighth, administrators need to continuously improve password policies by periodically monitoring their effectiveness, assessing user engagement to ensure users' awareness and compliance with existing policies and procedures, and evaluating the organization's policies against best practices and competitor organizations to ensure the robustness of the organization's policies.

Naqvi and Seffah [13] investigated how to align usability and security requirements in the early stages of the system development lifecycle. The authors also proposed a novel method and design patterns to ensure reuse. To ensure the effectiveness of the proposed method, it was evaluated across different domains including information systems, commercial off-the-shelf software, and cloud infrastructures. To identify conflicts between usability and security, the authors evaluated each system, off-the-shelf software package, and cloud infrastructure using the following criteria: confidentiality, integrity, availability, and nonrepudiation for security; and effectiveness, efficiency, and satisfaction for usability. The proposed method included the following steps: identify software requirements specification, list the security features or focus areas, elicit the security concerns, identify the usability versus security concerns, prioritize features, elicit the trade-off between usability and security, and document the trade-offs as reusable security design patterns. Implementing the proposed method can assist in creating a standardized catalog of usable security patterns and, hence, address conflicts between usability and security.

Aljedaani et al. [31] investigated the security awareness of end-users of mobile health (m-health) apps that are available on Android and iOS platforms. The authors aimed to provide guidelines for developers to be able to develop a secure and usable m-health app. The research-ers distributed a survey to collect users' feedback about two Saudi Arabia m-health providers. A total of 101 end users completed the survey. The survey investigated the security awareness of the users in terms of current and desired security features, security-related concerns, and methods to enhance security knowledge. The results of the survey indicated that end-users in Saudi Arabia were aware of the existing security features regardless of their sex, age, and edu-cational level. However, the end-users expressed a desire to improve the usability of security (e.g., biometric authentication) and stated some privacy concerns (e.g., data anonymization). The end-users also suggested enhancing the security knowledge broadcasting method. Increas-ing the security awareness of end-users can be achieved through social media and additional training.

Albesher [6] investigated the authentication processes of 20 websites to evaluate the usabil-ity by identifying authentication process issues during sign-up, sign-in, and password recovery and providing recommendations to enhance the process. The author utilized the "think-aloud" technique to analyze collected data more accurately. To achieve this, the author con-ducted a screen recording for each checked website. The results of this paper provided a list of recommendations to improve the authentication process and best practices for password rules and two-factor authentication. These rules and best practices can improve password reset pro-cesses and, hence, keep users' accounts secure.

Yu et al. [32] investigated guessing methods to compromise users' passwords by classifying the methods into trawling and targeted guessing. The authors also provided a large benchmark dataset that has been used for password guessing tasks and comprehensively analyzed all meth-ods to guess passwords. To achieve this, the authors re-viewed a set of benchmark datasets, analyzed development trends and their impact on the field, and evaluated several password-guessing methods to identify the effectiveness of each method. The result of this research showed that neural-network-based methods can increase the accuracy of guessing passwords. Analyzing these methods can assist administrators in setting password policies to make pass-words more difficult to guess.

Alroomi and Li [33] developed an automated method to assess the password creation poli-cies of websites by trying multiple passwords when a user signs up. The authors used multiple ML classifiers and heuristics to recognize and analyze signup forms and determine signup suc-cess. The developed method is considered the first method that can be applied to large-scale measurement, as it evaluates the creation policies of more than 20,000 websites. This offers sys-tem administrators a benchmark dataset to analyze password policy and, thus, assist them in creating stronger password policies to improve security. The results of the paper showed that many websites have few requirements for passwords, and more than half allow users to have passwords with six characters or less. This indicates that many websites accept weak pass-words. The results also showed that 12% of sites do not require minimum length requirements, 30% of sites do not support specific characters that are recommended in passwords, and only 12% of sites apply password blocklists.

## C. Password meters

Password meters assist users in adhering to password policies and creating secure passwords. They provide users with feedback regarding the strength of their newly created passwords [34]. These meters indicate the degree of password strength using various words and colors. To indicate a password's security level, some meters employ three colors: red for weak, orange

for fair, and green for strong. Some meters describe issues with generated passwords and offer solutions [35]. Other meters display a list of password composition criteria; when a user meets a criterion's requirements, the criterion's color changes or a green check mark appears next to it, assisting users in identifying any remaining issues.

Albesher [6] evaluated the authentication process for 20 leading websites in various domains (reservations, social media, shopping, entertainment, and tech services). The results showed that Facebook, Amazon, HUAWEI, and Microsoft do not use password meters. The results also showed that the meter works differently on the leading websites: either while the user types or after attempting to move to the following field or clicking the button for the next step.

Heuristics are used by password meters to direct users toward strong passwords. Certain heuristics anticipate user input [36]. To determine how strong a newly created password is, additional heuristics compare it to the passwords of other users [37, 38]. However, programming meters is difficult and can lead to inaccurate results. Albesher [6] found that the meter of Airbnb.com generated incorrect results.

Some studies found that password meters helped users to create stronger passwords while other studies found this information to be inaccurate [39]. Egelman et al. [40] found that the password meter helps to create a stronger password when the password meter is shown in the case of changing passwords but not in the case of creating passwords for the first time. Some researchers found that password meters annoyed users and made the task of creating passwords complicated [41, 42].

Password meters use length and entropy as the two factors that measure the difficulty of guessing a password. These two factors increase the time to crack a password [43]. Although password meters help users to satisfy these two factors through simplified text and colored presentation, studies [44] have shown that most users want to achieve the lowest level of password security permissible to be able to complete registration quickly.

Bojato et al. [45] adopted Password Guessability as a Service (PGaaS), which is a cloud-based service that assesses the strength of users' passwords based on a synthetic dataset generator and strength estimator. This service was implemented through Security as a Service (SaaS). The synthetic dataset generator relies on the generative adversarial network (GAN). This service assists users to ensure that their passwords are not easy to guess, identifies vulnerabilities within passwords, and, hence, reduces the possibility of their passwords being compromised. The authors developed PGaaS as a web and mobile service to estimate password strength and the time required to be guessed. The results of this paper show that PGaaS can enhance efficiency as it can detect vulnerable and non-vulnerable passwords and, hence, can assist users to select strong passwords and minimize vulnerabilities from malicious attacks.

Kariryaa and Schoning [46] developed MoiPrivacy, which is a password meter that utilizes a neural network and a heuristic-based approach to consider the personal information of a user when determining password strength and providing proper feedback. The developed meter was proposed to increase the awareness of the impact of using personal information in passwords. The developed meter analyzed the personal information of users in passwords through an online survey. This approach assists users in avoiding using personal information to create strong passwords and reduces the possibility of passwords being guessed. MoiPrivacy has been implemented as a browser extension to collect data and evaluate inserted passwords based on user data. The results of this paper show that MoiPrivacy can provide positive results by limiting users' ability to include personal information in passwords. This shows that Moi-Privacy can enhance online security by taking into consideration the personal information of users to create secure passwords.

Darbutaitė et al. [47] proposed a machine-learning approach to evaluate the strength of Lithuanian passwords. To start, the authors produced a dataset to evaluate the strength of the passwords. The proposed approach integrates English and Lithuanian language passwords to evaluate their strength based on 6 common features and 36 similarity metrics. The proposed method sets the Lithuanian language requirements to create strong passwords. This indicates that different languages need to have different meters that consider common words, phrases, and language vulnerabilities. The proposed method was able to predict the password strength with 77% accuracy. By considering the complexity of the Lithuanian language, the results of this paper are considered sufficient due to the low number of password-cracking tools targeting the Lithuanian language. The findings can also help to improve the estimation of password strength for Lithuanian users.

Kim et al. [48] developed a password meter system that hides the password exposure of user-chosen passwords by relying on fully homomorphic encryption (FHE) to ensure secure evaluation. Current password meters may expose previous user-chosen passwords. Therefore, the developed system allows the evaluation of users' passwords without exposing them and, hence, enhances security and privacy. This technique assists users in hiding their inserted passwords from being exposed to the password meter. This indicates that the developed password meter can provide a high level of security by ensuring confidentiality and privacy of user-chosen passwords. The developed system also encompasses contextual information in the password meter to assist users in selecting stronger passwords. The results of this paper show that the developed password meter can evaluate user-chosen passwords without exposing them but requires 60 seconds to evaluate due to hardware limitations.

Amador et al. [49] investigated the impact of prospect theory to increase password strength by showing weak user-chosen passwords compared to stronger ones. This leads users to select stronger passwords. This approach was evaluated by 762 participants and improved user-chosen password strength by 25%. Similarly, the evaluated approach was able to reduce user-chosen weak passwords by 25%. This shows that prospect theory employment can assist users by reducing the selection of weak passwords and, hence, increase overall security. To develop a proper password selection interface that aims to assist users in selecting stronger passwords, this study analyzed the relationship between given feedback and password decisions and between user behavior and mental models. These results can guide developers to improve the mechanisms that will be used to evaluate user-chosen passwords, aiming to guide users into creating stronger passwords.

Taneski et al. [50] analyzed the impact of a proactive password checker by using Markov models to evaluate the strength of passwords and realize password security. Several studies suggest that multiple Markov models are essential to calculate the strength of given passwords more accurately. This study investigated multiple models, aiming to identify the impact of different models for accurate predictions. The authors also investigated the impact of training password datasets on Markov models. The results of this study confirmed the need for multiple Markov models to accurately calculate password strength. The results also showed that training password datasets have a great impact on the Markov models' ability to effectively calculate the password strength.

## D. Password managers

Passwords are considered the most common method to authenticate users. However, millions of passwords have been compromised recently due to users choosing passwords based on their personal information, writing passwords down, reusing the same password across multiple sites, sharing passwords, storing passwords in plain text, or choosing easy-to-guess passwords

[51–53]. Password managers (PMs) address several security threats and usability issues to meet different requirements of different password policies by preventing users from memorizing the strong passwords of their accounts [51]. PMs are useful software applications that store generated and inserted passwords and manage passwords, aiming to assist users to access their accounts with minimal cognitive load. PMs play a critical role in ensuring password security while maintaining usability. Several studies have shown that memorizing strong passwords that meet the requirements of different password policies is difficult for users [54–57]. Therefore, security experts strongly suggest users utilize PMs to ensure password security [55–57]. However, several studies show that PMs have not been adopted widely due to users not trusting them, being unaware of the security advantages of them, viewing them as potential security risks, or usability issues [51, 54, 55].

Current PMs can be classified as browser-based, cloud-based, local-based, hardware-based, or system-wide and consist of several features [58, 59]. PMs can assist users in generating unique strong passwords for multiple accounts, simplify password management by required the user to remember only one master password, automatically fill in users' credentials, differentiate between legitimate and fraudulent websites, and identify weak or compromised passwords and, hence, ask users to change them [54, 55, 56]. This shows that PMs can protect users from common threats that aim to compromise users' passwords. The stored passwords are secured by a strong master encryption key that is hard to guess or brute-force [60].

Alodhyani et al. [51] studied the reasons for the low adoption of PMs by evaluating three different PMs from the perspectives of users who do and do not use PMs. To ensure valid results, the study included users who do use PMs as well as both expert and non-expert users who utilize PMs. The authors investigated the concerns through usability tests, interviews, and questionnaires to come up with valid recommendations that can address users' concerns. The results of this paper show that usability is not a big concern in comparison to other factors. Instead, the results show that the lack of trust and transparency issues are the main reasons for users not using PMs. Users have trust and security concerns regarding PM tools, which impact their decision to utilize PMs, while there are few concerns regarding user interfaces and the functionality of PMs. This emphasizes the necessity for developers to address these issues to ensure the high adoption of PMs.

Similar to Alodhyani et al. [51], Alshahrani and Alghamdi [56] investigated the critical factors that affect users' decisions to utilize PMs. The authors also discussed the positive impact of using PMs and their usefulness based on factors from the technology acceptance model and other factors from related work. To conduct this investigation, the authors relied on questionnaires to retrieve users' perspectives. The results of this study showed that ease of use, usefulness, user awareness, and user readiness have a great influence on users' decisions to utilize PMs. This indicates that PM developers must consider these factors to ensure the high adoption of PMs.

Dhanalakshmi et al. [53] investigated how to develop PMs that can store and encrypt passwords and other data securely. Despite the recent increase in data breaches, 77% of organizations do not have an incident response plan, which puts these organizations at high risk. Therefore, it is essential to understand how to store and generate secure passwords. One way to address this issue is to utilize a PM. To ensure a secure PM, the authors encompassed multi-factor authentication to access stored passwords. The proposed PM relies on a physical security key and a graphical user authentication mechanism to protect user data. The results of this paper can enhance the security of data stored in PMs by utilizing multi-factor authentication to ensure the validity of legitimate users.

Stobert et al. [54] proposed the ByPass PM to enhance usability by allowing direct communication between users and targeted websites for authentication purposes, reducing errors,

and simplifying password management tasks aiming to enhance security. The authors introduced the utilization of APIs within the PM to handle several tasks. The proposed PM reduces the required tasks that must be done by users to manage passwords and, hence, simplifies password management. ByPass addresses the issues of low adoption of PMs by allowing users to manage their accounts securely with low error rates. The results of this paper can address usability issues with better password management, help to manage accounts securely, minimize errors, and minimize required actions from users.

To address the risk issues when relying on a single master password to expose all passwords stored within a PM, Jeong and Jung [57] proposed the MonoPass PM that does not rely on a master password for authentication. Instead, it utilizes the master password to regenerate consistent passwords and passes password metadata to a central server for synchronization across devices while not storing user data on the server. The proposed PM can reduce the risks of all passwords of a user being compromised due to threats or security breaches of the PM and allows users to synchronize passwords across devices without storing user data on the server.

## Section 4: Materials & methods

The major aim of this study is to compare app authentication procedures in the e-commerce domain. The comparison focuses on the usability of these procedures. The term "usability" describes how well a user interacts with systems or products such as webpages, software, hardware, or apps. Several methods help researchers to evaluate the usability of app authentication procedures. One of these methods is usability inspection [61]. The inspection type that was selected is called "individual expert review," which is a user-centered design method that does need user involvement [62]. A single expert examines an interface for potential issues occasionally using tasks and guidelines but frequently informally to identify potential problems based on their personal experience [63]. Several previous studies have used this methodology [6, 64–66].

The researchers of this study examined the sign-up, sign-in, and password recovery processes for 50 different apps. The comparison of a set of items for three different authentication processes makes this study comprehensive. During the sign-up process, the researchers checked if the app asked the user to retype the password, allowed hiding or showing the password, showed a password meter, imposed certain password rules, permitted the use of a password manager, and verified the entered email address or phone number. The items checked for the sign-in process include whether the app asked for the email address and/or password to be re-entered after every failed attempt, whether it granted the user the option of staying signed in and saving the password, and whether it used biometric authentication. The items for the recovery process are the same as the registration process, in addition to checking if the app applied a robot check and whether it used the OTP option. Every app was examined in a different session, which was recorded using the iPhone's screen recorder. The examiner followed the think-aloud technique and, thus, they made sure the microphone was on in every session. As part of the think-aloud research technique, the examiner speaks the thoughts that come to his/her mind. Charters [67] stated that the think-aloud technique is one of the most effective ways to assess higher-level thinking processes.

The authors selected the e-commerce domain for several reasons. The first is that e-commerce can be accessed by anyone around the world, unlike some apps such as bank apps. The second reason is that e-commerce apps face a lot of attacks because it is a targeted domain. The third reason is that many people save their personal and banking information in these apps. The 50 apps were selected randomly from the App Store.

## Section 5: Results

The detailed results of the items checked during the sign-up process are shown in Table 1. Only 9 out of the 50 apps asked users to re-enter their password. Furthermore, the majority of the apps (41) allowed users to view passwords as they typed. Additionally, 41 of the apps required the user to follow certain rules while setting up new passwords. The results also revealed that only 6 out of 50 apps offered a password meter. Moreover, all apps asked users to provide an email address and/or phone number for verification.

Table 2 shows the details of the password rules enforced by every app. The password rules for Ounass, eBay, and Amazon were displayed before the user entered the password, while the policy for Level Shoes appeared once the user had started typing. The most common limit for password characters is eight. However, some apps have different limits. For example, Asos required a minimum of ten characters, while Farfetch asked for a minimum of only four characters. No additional rules were presented in the apps when signing up. However, some apps have rules that are not discovered unless the user breaks them.

Table 3 shows the differences between meters. The meters used either red or orange colors for weak passwords and green colors for strong ones. In certain apps, such as Centerpoint, the meter operated only when typing. In the other apps, the meter only operated after moving to the next field or pressing the button for the next step. Fifteen different apps (Bath & Body Work, Mothercare, Sephora, Next, H&M, Adidas, IKEA, Massimo Dutti, Zara Home, Bloomingdales, Lefties, American Eagle, Stradivarius, Jarir, and eBay) had individual rules shown on separate crossed lines with colors changing to indicate adherence. Certain apps employed either the color green or a crossed line to indicate when rules were met.

According to the results of the sign-in procedure (Table 4), all apps asked users to retype their phone numbers/emails and passwords after every failed attempt. A total of 36 out of 50 apps allowed customers to show or hide passwords as they type. The "stay signed in" feature was available for all apps. Additionally, all apps allowed the use of password managers to save passwords. Moreover, 12 apps did not use biometrics as a method of authentication.

Table 5 displays the results of the checked items during the recovery process. A total of 35 apps adhered to password rules during account recovery. Only 19 apps showed password

**Table 1. The results of the sign-up process.**

| App | Retyping the password | Hiding or showing the password | Password policies | Password meter | Verifying email or phone |
|---|---|---|---|---|---|
| Level Shoes, eBay | √ | √ | √ | X | √ |
| iHerb | X | √ | √ | √ | √ |
| SIVVI | X | √ | X | X | √ |
| Nice One, Oysho | X | √ | X | X | √ |
| Vogacloset, RIVA | √ | √ | X | X | √ |
| Amazon, Asos 10, Bath & Body Work, Mothercare, Farfetch, Sephora, Next, H&M, Ounass, The Outnet, AliExpress, Nick, Zara, Adidas, Bershka, IKEA, Massimo Dutti, Zara Home, Bloomingdales, Lefties, American Eagle, Styli, NAMSHI, Golden Scent, YOOX | X | √ | √ | X | √ |
| Childrensalon, Mamas World, Femi9 | X | X | X | X | √ |
| Home Box, Centerpoint, Baby Shop | X | √ | √ | √ | √ |
| Noon, MAX | X | √ | √ | √ | √ |
| Charles & Keith | √ | X | √ | X | √ |
| Stradivarius, Mango, GAP, The Beauty Secrets | X | X | √ | X | √ |
| Nejree | √ | X | X | X | √ |
| Saks Fifth, Jarir, Faces | √ | √ | √ | X | √ |

**Table 2. The password rules for each of the test apps.**

| App | Password rules |
|---|---|
| Amazon, Ounass, Home Box, Centrepoint, Baby Shop, Style, NAMSHI, YOOX | At least 6 characters. |
| Asos | Minimum 10 characters. |
| Bath & Body Works | The rules appeared after clicking on the box; at least 8 characters (letters + 1 number + 1 special character space). |
| Mothercare | The rules appeared after clicking on the box; at least 7 characters (letters + 1 number + 1 special character space). |
| Farfetch | No rule until user clicks "register"; at least 4 characters. |
| Sephora | Minimum 8 characters (lowercase letters, uppercase letters, and numbers). |
| Next | The rules appeared after clicking on the box; 6–12 characters + 1 number. |
| H&M | At least 8 characters (letters + 1 number + 1 special character, no spaces). |
| The Outnet | Minimum 8 characters (uppercase letter + number). |
| AliExpress | The rules appeared after clicking on the box; 6–20 characters (number and letters only). |
| Zara | Minimum 8 characters (lowercase letters, uppercase letters, and numbers). |
| Adidas | The rule appeared after clicking on the box; 8 characters (1 lowercase letter, 1 uppercase letter, 1 number + 1 special character). |
| Bershka | Minimum 8 characters (lowercase letters, uppercase letters, and numbers). Cannot repeat the same character more than three times. |
| IKEA | 8–20 characters (lowercase letters, uppercase letters, and numbers or special characters). No more than two identical characters in a row. |
| Massimo Dutti | Minimum 8 characters (lowercase letters, uppercase letters, and numbers). Cannot repeat the same character. |
| Zara Home | Maximum 8 characters (lowercase letters, uppercase letters, and numbers). Cannot repeat the same letter. |
| Bloomingdales | The rules appeared after starting writing; a minimum of 8 characters including both an uppercase letter and a number. Cannot use space or (., —\| / =). |
| Lefties | Minimum 8 characters (1 lowercase letter, 1 uppercase letter, and 1 number). Cannot repeat characters. |
| American Eagle | The rules appeared after clicking on the box; at least 7 characters (1 number + 1 special character or space). |
| Golden Scent | At least 7 characters (One uppercase character). |
| Noon, Faces, iHerb | At least 8 characters. |
| MAX, Nick | No rules. |
| Charles & Keith | The rule appeared after starting writing; at least 8 characters consisting of upper- and lower-case letters with at least 1 number. |
| Stradivarius | Minimum 8 characters (1 uppercase letter and 1 number). |
| Mango | The rules appeared after clicking on the box; 10–30 characters with at least 1 letter and 1 number. |
| GAP | 8–20 characters contain at least one numeric digit, uppercase, and lowercase letter. |
| The Beauty Secrets | The registration button does not activate until writing a minimum of 3 symbols or an invalid character. |
| Saks Fifth | The rule appeared after starting to write at least 8 characters (uppercase letters, lowercase letters + symbol). |

(*Continued*)

**Table 2.** (Continued)

| App | Password rules |
|---|---|
| Jarir | At least 8 letters (1 lowercase letter, 1 uppercase letter, 1 special character, and 1 number). |
| Level Shoes | The rules appeared after start writing; at least 8 characters. |
| eBay | At least 8 letters (1 number or symbol and 1 letter). |

meters. Twenty-one (21) apps asked for passwords to be retyped, while 29 apps allowed the hiding or showing of passwords. Only seven (7) apps used the OTP as a method of verifying users. The robot check was applied in only three (3) apps during recovery, while no apps applied it during sign-up. A total of 39 out of 50 apps used email links to change passwords. Although 14 apps requested users' phone numbers during the registration, they did not use them for password recovery. The Ounas app failed to deliver a password reset link to the email.

## Section 6: Discussion

This research aimed to find similarities and differences between security procedures in mobile apps. The findings demonstrated that there are major differences among apps in terms of security procedures. These differences create a heavy cognitive burden on users' memory [68–71]. One of the key differences is in the password rules. For example, some apps ask for a minimum of six characters for passwords while others ask for a minimum of eight characters. Another difference is in the functionality and presentation of password meters. This diversity underscores the importance of considering usability and user experience metrics when designing authentication processes. Striking the right balance between security and usability is an ongoing challenge for app developers [72–74]. The variations in authentication rules and tools among different apps indicate a lack of unified knowledge about what is best for users.

Some of the tested apps incorporate biometric authentication features such as face recognition. Biometric authentication has been shown to be an effective added method to compliment the use of passwords [75]. Recent developments in biometric technology have improved its accuracy and security in mobile apps, making it more reliable [5, 76, 77]. This development has increased the acceptance, speed, and effectiveness of biometric authentication [5], which leads to a usable security procedure. Many apps now prefer to use two-factor authentication to protect users' financial information, such as credit card details [78].

The results from this research indicate that 9 out of the 50 apps tested ask users to retype their passwords during registration. Although this practice is time-consuming, it contributes positively to both usability and security [79] by reducing the likelihood of typing errors [6], which in turn minimizes the need for password reset requests. The trade-off between security and usability is also found in the use of a "password manager" feature, which was found to be common in all the tested apps. This feature becomes highly risky when users reuse passwords

**Table 3.** The password meters for some apps.

| App | Password meter |
|---|---|
| iHerb | "very weak" in red, "weak" in orange, "good" & "great" in green |
| Home Box, Centrepoint, Baby Shop | "not eligible" and "weak" in orange, and "good" & "strong" in green |
| Noon | "weak" in red, "average" in orange, and "good" & "strong" in green |
| MAX | "not eligible" and "weak" in orange, and "good" & "strong" in green |

**Table 4. The results of the sign-in process.**

| App | Retype email after failed attempt | Retype password after failed attempt | Stay signed in | Use password manager | Hide or show password | Biometrics |
|---|---|---|---|---|---|---|
| ZARA, Level Shoes, RIVA, eBay, iHerb, AliExpress, Noon, Ounass, H&M, SIVVI, NAMSHI, Golden Scent, Farfetch, Asos, Amazon, IKEA, Mothercare, Bath & Body Work, Faces, Nick, Bershka, Mango, Zara Home, Bloomingdales, American Eagle, Oysho, Saks Fifth, Home Box | √ | √ | √ | √ | √ | √ |
| Centerpoint, The Outnet, MAX, Vogacloset, Sephora, Baby Shop | √ | √ | √ | √ | √ | X |
| Next, YOOX, Nice One, Stradivarius, Massimo Dutti, Lefties, Jarir, Ejree | √ | √ | √ | √ | X | √ |
| Childrensalon, Mamas World, Femi9, The Beauty Secrets, GAP, Charles & Keith | √ | √ | √ | √ | X | X |

across multiple accounts, which potentially increases the possibility of attack [80]. The practice of reusing passwords was observed in 44 apps out of 50. Mayer et al. [81] found that 77% of 277 participants in their study reused passwords for several accounts, while Poornachandran et al. [82] found that 59% of 1,508 participants in their survey disclosed password reusage. Users tend to reuse a password because it is memorable even though they know it affects the security of their accounts [83]. The issue of memorability is one of the most common issues for some unacceptable security practices [84]. In the case of reusing passwords, people valued usability more than security because they were afraid of resetting passwords regularly.

Our study revealed that password rules varied from one app to another. Password rules ensure secure password usage, typically necessitating the creation of complex and unique passwords for each system. For instance, some apps disallowed the use of the last password used,

**Table 5. The results of the recovery process.**

| App | Password policies | Password meter | Retyping password | Hide or show password | OTP | Robot check |
|---|---|---|---|---|---|---|
| ZARA, Next, Farfetch, Sephora, Adidas, Massimo Dytti, Zara Home, Bloomingdales, American Eagle | √ | X | X | √ | X | X |
| Level Shoes, AliExpress | √ | X | √ | √ | √ | X |
| RIVA | X | X | √ | X | √ | X |
| Centerpoint, Baby Shop, Home Box, MAX, Mothercare, Bath & Body Work | √ | √ | X | √ | X | X |
| eBay, Amazon, Nike | √ | X | X | √ | √ | X |
| iHerb, Ounass, Faces | X | X | X | X | X | X |
| The Outnet | √ | √ | X | √ | X | √ |
| Noon, SIVVI, IKEA | X | X | X | X | X | X |
| H&M | √ | X | X | √ | X | X |
| NAMSHI, Jarir, Lefties, GAP, Charles & Keith | √ | X | √ | X | X | X |
| Golden Scent, YOOX, Saks Fifth, Asos | √ | X | √ | √ | X | X |
| NICE ONE, Vogacloset | X | X | √ | √ | X | X |
| Childrensalon, Mamas World, Nejree | X | X | √ | X | X | X |
| Femi9 | X | √ | √ | X | X | X |
| The Beauty Secrets | √ | √ | √ | X | X | √ |
| Styli | X | √ | √ | √ | X | X |
| Bershka | √ | X | √ | X | X | X |
| Stradivarius | √ | X | X | X | √ | √ |
| Mango | √ | X | X | X | X | X |
| Oysho | X | X | X | √ | X | X |

repeating numbers or characters more than three times, and the use of spaces in passwords. Research has shown that individuals tend to choose weaker and more memorable passwords when encountering complex password rules [85]. Furthermore, individuals may respond to password rules by reusing passwords across several apps or leaving written passwords in visible places [86, 87]. The results also indicated that 41 out of the 50 apps used password meters to indicate the strength of the password during registration. These meters varied when they provided feedback to users since some assessed password strength as the user typed while others displayed the assessment after the user had finished typing the password. Password meters play a crucial role in guiding users toward choosing stronger and more secure passwords [88].

Our study showed that 2 out of the 50 apps (i.e., Ounass and Centerpoint) are late in sending links for the recovery process. Innocenti et al. [89] analyzed the resetting process that uses email links and demonstrated that a significant number of systems are vulnerable to concrete password-recovery reset attacks, which have the potential to result in account takeover. In particular, their study found that 163 (45%) of the tested websites failed to notify the account owner of password change requests. Our research revealed that 6 out of the 50 apps tested allowed users to reset their passwords by sending them an OTP to their cell phone. In certain situations, such as when traveling without enabling international messaging, the user would not be able to receive OTPs.

As e-commerce increasingly relies on mobile apps to reach customers, the authentication of these apps becomes a critical point of vulnerability. It is essential to balance ease of use with the highest level of security, but the methods to achieve this are not without their trade-offs. On one hand, traditional passwords and PINs provide a familiar experience for users. They are already understood, and implementation is relatively straightforward for developers. However, they are notoriously weak. Passwords are often reused, easy to guess, or ultimately cracked by determined attackers. PINs, while slightly more secure due to being numeric, still fall short. On the other hand, more modern biometric methods like facial or fingerprint recognition offer vastly improved security. They are much harder for attackers to replicate fraudulently. Yet, they can also introduce friction for the user. Compatibility issues between devices can arise, and some users may be uncomfortable with the privacy implications of storing such sensitive personal data. Another avenue is multi-factor authentication (MFA). By requiring both a password (something you know) and a second form of verification (something you have, like a phone), MFA greatly enhances security. One-time codes sent via SMS do this. While better, this method is still vulnerable to sophisticated sim-swapping attacks. Authenticator apps, which generate codes but store no sensitive data server-side, are a slight improvement. Ultimately, the best approach will depend on the specific risk tolerance of the e-commerce service and the preferences of its target audience. There is no one-size-fits-all solution to balancing security and usability for mobile app authentication. However, by carefully weighing the options and potentially offering choices to the user, it is possible to create an experience that is both secure and accessible. As threats continue to evolve, so too must the methods used to protect users.

The study eventually centers on the trade-offs between security, resilience, user experience, and complexity. While unified or centralized systems offer simplicity and potential efficiency gains, they also introduce significant security risks. On the other hand, distributed models provide greater security and resilience but require more complex design and implementation to ensure a positive user experience. The optimal approach likely lies in a balanced hybrid model that leverages the strengths of both centralized and decentralized architectures. For example, a system could use centralized elements for streamlined credential management, while incorporating decentralized components to add layers of security and reduce single points of failure.

Ultimately, the specific requirements and constraints of the use case will need to guide the design decisions.

## Section 7: Future research directions

The study showed clearly that apps differ in their design for authentication procedures. These differences add a burden to users' memory and understanding, while no app can guarantee that they follow best practices that are based on the knowledge of usability and security. This study clarifies the need for unified usable procedures for authentication that ensure metrics of both usability and security. Usability designers should put more effort into generating a framework that can help app developers apply more usable authentication procedures. One study method that can help to detect what procedures are more usable and secure is conducting experiments with users. These experiments could test the efficiency, effectiveness, and users' satisfaction with certain procedures.

The current research only covered mobile apps in the mobile commerce field. In this sector, the developers of the apps may prioritize usability over security to gain more customers and allow them to do transactions quickly without any interruption with any security procedures. On the other hand, the developers of apps in the health sector may prioritize security and usability even if the procedures take more time and frustrate users. Therefore, it is important to conduct a study that compares apps from different fields.

## Section 8: Conclusions

The use of smartphone apps is growing rapidly. Therefore, developers must increase the usability and security of apps. This paper aimed to find out to what extent there are usable security authentication practices in mobile apps. In addition, the results of this paper will help developers choose the most common authentication practices for apps to ensure usability and security for users. The findings of this paper should enhance the usability and security of smartphone apps. Specifically, this study evaluated several authentication practices by linking their usage to the findings of other studies. This evaluation should guide improving the usable security of mobile apps.

## Author Contributions

**Conceptualization:** Abdulmohsen Saud Albesher, Amal Alkhaldi.

**Data curation:** Abdulmohsen Saud Albesher.

**Formal analysis:** Abdulmohsen Saud Albesher, Ahmed Aljughaiman.

**Funding acquisition:** Abdulmohsen Saud Albesher.

**Investigation:** Abdulmohsen Saud Albesher, Ahmed Aljughaiman.

**Methodology:** Abdulmohsen Saud Albesher, Amal Alkhaldi.

**Project administration:** Abdulmohsen Saud Albesher.

**Resources:** Abdulmohsen Saud Albesher, Amal Alkhaldi, Ahmed Aljughaiman.

**Supervision:** Abdulmohsen Saud Albesher.

**Validation:** Abdulmohsen Saud Albesher, Ahmed Aljughaiman.

**Visualization:** Abdulmohsen Saud Albesher, Amal Alkhaldi.

**Writing – original draft:** Abdulmohsen Saud Albesher, Amal Alkhaldi.

**Writing – review & editing:** Abdulmohsen Saud Albesher, Ahmed Aljughaiman.

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
