## [Decision Letter · Decision Letter 0]

5 Nov 2024

PONE-D-24-44697Toward Secure Mobile Applications Through Proper Authentication MechanismsPLOS ONE

Dear Dr. Albesher,

Thank you for submitting your manuscript to PLOS ONE. After careful consideration, we feel that it has merit but does not fully meet PLOS ONE’s publication criteria as it currently stands. Therefore, we invite you to submit a revised version of the manuscript that addresses the points raised during the review process.

Minor Revision required.

We look forward to receiving your revised manuscript.

Kind regards,

Elochukwu Ukwandu, PhD

Academic Editor

PLOS ONE

Journal Requirements:

 This research paper was funded by the Deanship of Scientific Research, Vice Presidency for Graduate Studies and Scientific Research, King Faisal University, Saudi Arabia under the [GRANT No. 4215].

The authors extend their appreciation to the Deanship of Scientific Research, Vice Presidency for Graduate Studies and Scientific Research, King Faisal University, Saudi Arabia [GRANT No. 4215].

 This research paper was funded by the Deanship of Scientific Research, Vice Presidency for Graduate Studies and Scientific Research, King Faisal University, Saudi Arabia under the [GRANT No. 4215].

Reviewers' comments:

Reviewer's Responses to Questions

**Comments to the Author**

1. Is the manuscript technically sound, and do the data support the conclusions?

Reviewer #1: Yes

Reviewer #2: Yes

2. Has the statistical analysis been performed appropriately and rigorously? 

Reviewer #1: N/A

Reviewer #2: Yes

3. Have the authors made all data underlying the findings in their manuscript fully available?

Reviewer #1: Yes

Reviewer #2: Yes

4. Is the manuscript presented in an intelligible fashion and written in standard English?

Reviewer #1: Yes

Reviewer #2: Yes

5. Review Comments to the Author

Reviewer #1: Include a section on research directions in the paper. Add a reference to the paper 'CMAF-IIoT: Chaotic Map-based Authentication Framework for Industrial Internet of Things' in the text. Insert an abbreviation table in the revised manuscript. Additionally, include a few lines discussing the significance of the paper

Reviewer #2: The work is highly relevant to the current landscape, where the rapid rise of mobile apps has brought their integrity, confidentiality, and overall security into sharp focus. With mobile applications transitioning from a luxury to an essential part of daily life, concerns around the protection of sensitive user data are more pressing than ever. As apps increasingly handle personal and financial information, the need for robust security mechanisms is undeniable.

The suggestion of a unified and user-friendly authentication procedure, aimed at reducing the cognitive burden of managing multiple passwords, is a logical approach. Simplifying the login process would enhance user experience and lower the barriers to app adoption. However, this introduces a critical debate: while centralized systems for password management might streamline access, they inherently create single points of failure. In contrast, distributed authority models are seen as more secure, offering resilience by decentralizing control. How does the author debate on this point is the main concern? The paper would be complete if this point is touched upon and debated. These models mitigate the risks of mass data breaches but may come with their own challenges, such as increased complexity in managing security protocols across multiple nodes.

Additionally, the ever-evolving threat landscape, with cyber-attacks growing more sophisticated, makes it imperative to carefully evaluate the strengths and limitations of each approach. It is essential to balance ease of use with the highest level of security. This needs to be justified.

All in all, this paper presents a well-researched, thorough survey of the authentication methods employed by mobile apps today, shedding light on potential future directions and improvements in mobile app security. The exploration of the trade-offs between usability and security, as well as centralized versus distributed control, adds significant value to ongoing discussions in the field.

6. PLOS authors have the option to publish the peer review history of their article (what does this mean?). If published, this will include your full peer review and any attached files.

Reviewer #1: No

Reviewer #2: No

---

## [Author Response · Author response to Decision Letter 0]

15 Nov 2024

Reviewer #1: 

1- Include a section on research directions in the paper. 

Answer: Thanks for this comment. We created a new section (Future Research Directions). We moved the last paragraph of the conclusion section and added a new paragraph to this new section.

2- Add a reference to the paper 'CMAF-IIoT: Chaotic Map-based Authentication Framework for Industrial Internet of Things' in the text. 

Answer: Thanks for this comment. It is an interesting research paper which is related to our work. We added it and talked about it in the last section of Related Work (A).

3- Insert an abbreviation table in the revised manuscript. 

Answer: Thanks for your comment. We agree that we mentioned a lot of abbreviations and thus it might be better to move their detentions to a table. You can find the added table before the references section.

4- include a few lines discussing the significance of the paper

Answer: Thanks for this comment. We added more information about the significance of our paper before the last paragraph of the introduction.

Reviewer #2: 

1- The work is highly relevant to the current landscape, where the rapid rise of mobile apps has brought their integrity, confidentiality, and overall security into sharp focus. With mobile applications transitioning from a luxury to an essential part of daily life, concerns around the protection of sensitive user data are more pressing than ever. As apps increasingly handle personal and financial information, the need for robust security mechanisms is undeniable.

Answer: Thanks for this comment. We are glad that an expert highlights the importance of our study.

2- The suggestion of a unified and user-friendly authentication procedure, aimed at reducing the cognitive burden of managing multiple passwords, is a logical approach. Simplifying the login process would enhance user experience and lower the barriers to app adoption. However, this introduces a critical debate: while centralized systems for password management might streamline access, they inherently create single points of failure. In contrast, distributed authority models are seen as more secure, offering resilience by decentralizing control. How does the author debate on this point is the main concern? The paper would be complete if this point is touched upon and debated. These models mitigate the risks of mass data breaches but may come with their own challenges, such as increased complexity in managing security protocols across multiple nodes.

Answer: Thanks for this comment. It adds value to our article because it is related to one solution that minimizes cognitive load (i.e. unified authentication). We added a paragraph to show this debate at the end of the discussion section.

3- Additionally, the ever-evolving threat landscape, with cyber-attacks growing more sophisticated, makes it imperative to carefully evaluate the strengths and limitations of each approach. It is essential to balance ease of use with the highest level of security. This needs to be justified.

Answer: Thanks for this comment. We added one paragraph to address this issue. The paragraph starts with “As e-commerce…” and is located at the end of the discussion section.

4- All in all, this paper presents a well-researched, thorough survey of the authentication methods employed by mobile apps today, shedding light on potential future directions and improvements in mobile app security. The exploration of the trade-offs between usability and security, as well as centralized versus distributed control, adds significant value to ongoing discussions in the field.

Answer: Thanks for this comment. We added a paragraph about this at the end of the discussion section.

---

## [Decision Letter · Decision Letter 1]

22 Nov 2024

Toward Secure Mobile Applications Through Proper Authentication Mechanisms

PONE-D-24-44697R1

Dear Dr. Albesher,

We’re pleased to inform you that your manuscript has been judged scientifically suitable for publication and will be formally accepted for publication once it meets all outstanding technical requirements.

Kind regards,

Elochukwu Ukwandu, PhD

Academic Editor

PLOS ONE

Additional Editor Comments (optional):

Reviewers' comments:

Reviewer's Responses to Questions

**Comments to the Author**

1. If the authors have adequately addressed your comments raised in a previous round of review and you feel that this manuscript is now acceptable for publication, you may indicate that here to bypass the “Comments to the Author” section, enter your conflict of interest statement in the “Confidential to Editor” section, and submit your "Accept" recommendation.

Reviewer #1: All comments have been addressed

Reviewer #2: All comments have been addressed

2. Is the manuscript technically sound, and do the data support the conclusions?

Reviewer #1: Yes

Reviewer #2: Yes

3. Has the statistical analysis been performed appropriately and rigorously? 

Reviewer #1: Yes

Reviewer #2: Yes

4. Have the authors made all data underlying the findings in their manuscript fully available?

Reviewer #1: Yes

Reviewer #2: Yes

5. Is the manuscript presented in an intelligible fashion and written in standard English?

Reviewer #1: Yes

Reviewer #2: Yes

6. Review Comments to the Author

Reviewer #1: Authors address all the comments, the paper can be accepted. No further changes are required at this time

Reviewer #2: All the concerns raised by the reviewers has been addressed by the authors and the expectations has been met.

7. PLOS authors have the option to publish the peer review history of their article (what does this mean?). If published, this will include your full peer review and any attached files.

Reviewer #1: No

Reviewer #2: No

---

## [Editor Report · Acceptance letter]

26 Nov 2024

PONE-D-24-44697R1 

PLOS ONE

Dear Dr. Albesher, 

I'm pleased to inform you that your manuscript has been deemed suitable for publication in PLOS ONE. Congratulations! Your manuscript is now being handed over to our production team.

Kind regards, 

on behalf of

Dr. Elochukwu Ukwandu 

Academic Editor

PLOS ONE